:◌̈: PLOS | ONE

# Human umbilical cord blood monocytes, but not adult blood monocytes, rescue brain cells from hypoxic-ischemic injury: Mechanistic and therapeutic implications

**Arjun Saha**[ID]*, **Sachit Patel**[¤], **Li Xu, Paula Scotland, Jonathan Schwartzman**[ID], **Anthony J. Filiano, Joanne Kurtzberg, Andrew E. Balber**

Marcus Center for Cellular Cures (MC3), Duke University School of Medicine, Durham, North Carolina, United States of America

¤ Current address: Department of Pediatrics, Division of Pediatric Blood and Marrow Transplantation, University of Nebraska Medical Center, Omaha, Nebraska, United States of America
* arjun.saha@duke.edu

**Data Availability Statement:** All relevant data are within the paper and its Supporting Information files.

## Abstract

Cord blood (CB) mononuclear cells (MNC) are being tested in clinical trials to treat hypoxic-ischemic (HI) brain injuries. Although early results are encouraging, mechanisms underlying potential clinical benefits are not well understood. To explore these mechanisms further, we exposed mouse brain organotypic slice cultures to oxygen and glucose deprivation (OGD) and then treated the brain slices with cells from CB or adult peripheral blood (PB). We found that CB-MNCs protect neurons from OGD-induced death and reduced both microglial and astrocyte activation. PB-MNC failed to affect either outcome. The protective activities were largely mediated by factors secreted by CB-MNC, as direct cell-to-cell contact between the injured brain slices and CB cells was not essential. To determine if a specific subpopulation of CB-MNC are responsible for these protective activities, we depleted CB-MNC of various cell types and found that only removal of CB CD14$^+$ monocytes abolished neuroprotection. We also used positively selected subpopulations of CB-MNC and PB-MNC in this assay and demonstrated that purified CB-CD14$^+$ cells, but not CB-PB CD14$^+$ cells, efficiently protected neuronal cells from death and reduced glial activation following OGD. Gene expression microarray analysis demonstrated that compared to PB-CD14$^+$ monocytes, CB-CD14$^+$ monocytes over-expressed several secreted proteins with potential to protect neurons. Differential expression of five candidate effector molecules, chitinase 3-like protein-1, inhibin-A, interleukin-10, matrix metalloproteinase-9 and thrombospondin-1, were confirmed by western blotting, and immunofluorescence. These findings suggest that CD14$^+$ monocytes are a critical cell-type when treating HI with CB-MNC.

## Introduction

Mononuclear cell (MNC) prepared from human umbilical cord blood (CB) are candidate therapeutics for treating hypoxic-ischemic (HI) brain injuries. Patients with cerebral palsy [1–4],

**Funding:** This work was supported by grants from the Julian Robertson Foundation and the Marcus Foundation.

**Competing interests:** The authors have declared that no competing interests exist.

**Abbreviations:** CB, cord blood; HI, hypoxia-ischemia; HIE, hypoxic- ischemic encephalopathy; MNC, mononuclear cell; OGD, oxygen glucose deprivation; PB, peripheral blood.

neonatal hypoxic-ischemic encephalopathy (HIE) [5], and acute ischemic stroke [6] have been treated with intravenously administered CB-MNC in early safety and feasibility trials. Some signals of efficacy have emerged in a Phase 2 trial in young children with cerebral palsy [7], and additional clinical studies involving treatment of ischemic brain injury with CB are currently listed as open on ClinTrials.gov (stroke, NCT02433509,. NCT0167393, NCT0300497, NCT0143859, NCT02881970; neonatal hypoxic ischemic encephalopathy, NCT0243496, NCT02612155, NCT0225661, NCT0255100, NCT0228707; various conditions, NCT0332746).

Many preclinical studies suggest that CB-MNC protect the brain after HI by releasing neurotrophic and anti-inflammatory factors that stimulate repair by host cells [8, 9]. These studies, using various animal and culture systems, have implicated different CB-MNC subpopulations in contributing to neuroprotection [10–18]. CB-MNC protect primary astrocytes [19], oligodendrocytes [20, 21], and microglia [19], as well as neuronal cell lines [15] from HI-induced injury. However, what factors mediate brain repair, the CB-MNC cell types that contribute, and the host cells with which they interact are unclear. Determining which cell types in CB-MNC enhance brain tissue repair, and the mechanisms by which they do so, will optimize decisions on dosing, route of administration, treatment frequency, and other critical clinical and regulatory parameters. This information may also help in the development of mechanism-based potency assays for advanced clinical testing and, ultimately, for manufacturing and releasing products for clinical use.

In this paper, we present experiments that address these questions using organotypic mouse brain slice cultures exposed to oxygen-glucose deprivation (OGD) [22–25]. The brief OGD exposure triggers a neuro-inflammatory cascade involving activation of microglia and astrocytes that leads to the death of neurons over 2 to 3 days. Organotypic slice cultures offer the advantage of preserving the cytoarchitecture of the tissue of origin and connectivity of different anatomical regions, as well as functional relationships and interactions between neighboring cells, such as neurons and astrocytes, keeping the intrinsic synaptic connections found *in vivo*. Because the brain architecture and cell types are preserved in this model, the pathogenic mechanisms induced by OGD in brain slices are similar to those causing HI brain injuries *in vivo* [26]. Thus, this model can be used to test cell therapies for neuroprotection following OGD [20, 26]. We report here that CB-MNC protect neurons from death and dampen the activation of astrocyte and microglia in slice cultures exposed to OGD. This neuroprotection was mediated by CD14$^+$ monocytes in the CB-MNC. Unlike CB monocytes, CD14$^+$ monocytes from adult peripheral blood (PB) did not confer protection to neurons or reduced glial activation. We identified several candidates upregulated, at the RNA and protein levels, in CB monocytes compared to PB monocytes that may play a role in neuroprotection and repair. These findings will inform late stage clinical development of CB-MNC products for treatment of HI brain injury.

## Material and methods

### Animals

All experiments were performed in accordance with Duke University Institutional Animal Care and Use Committee's policies and followed approved protocols. The protocol was approved by the Duke IACUC (Protocol Number: A020-17-01) for all the animal related studies described in this paper including mouse organotypic brain slice culture. C57BL/6 mice (The Jackson Laboratory) and CX3CR1-GFP$^+$/- mice were maintained in Duke Facilities under direct veterinary supervision. Animals had ad libitum access to food and water in a temperature-controlled room under a 12-hour light: 12-hour dark illumination cycle.

## Oxygen-glucose deprivation (OGD) of brain slice cultures

Organotypic forebrain slice cultures were prepared following the method described by Stoppini *et al.*[22]. Briefly, 300μm thick forebrain sagittal slices from postnatal day-2 mouse pups, sacrificed by decapitation, were sectioned. The sections were cultured under controlled atmospheric conditions on top of cell impermeable membranes in contact with culture medium. Slices were exposed to medium without glucose in an oxygen-free gas mixture for one hour, returned to normoxic, glucose replete conditions, and incubated for 72 hours before further analysis.

## Treating slice cultures with cells

To test the protective activity of human CB or PB cell populations, $2.5 \times 10^4$ cells were added directly onto each brain slice immediately after OGD shock. Alternatively, to check possible paracrine effect, cell populations were added to the tissue culture medium below the membrane supporting slices. And in this case, to compensate for possible dilution of protective factors by the large volume the culture medium, we added $1.25 \times 10^5$ cells below the membrane. Cell death and/or the cellular composition of the slice cultures were compared to control slices not treated with cells 72 hours after OGD treatment.

## Evaluation of cell death following OGD

Slices were transferred to medium containing propidium iodide (PI, 2.0 μg/mL, Sigma) and incubated for 30 minutes to stain late apoptotic and necrotic cells, washed thoroughly with PBS, fixed with 4% paraformaldehyde [PFA] containing 4,6-diamidino-2-phenylindole (DAPI). Slides were coded, and percentage of total cells (DAPI staining) that were dead (PI staining) in multiple sequential images of the periventricular region was determined. To minimize regional variation, we kept the region of analysis uniform (periventricular region) between experimental groups. Each slide was analyzed by an investigator blinded to the identity of the experimental material using a Leica SP8 upright confocal microscopy (Leica Microsystems, IL, USA) and ImageJ and Plugin Cell Counter (NIH Image, USA) software.

## Immunohistological analysis of cell populations in slice cultures

Brain slices were fixed in 4% PFA and blocked in phosphate buffered saline (PBS) containing 3% heat-inactivated horse serum, 2% bovine serum albumin (BSA), and 0.25% triton-X-100 overnight. Primary antibody was prepared in 2% BSA, 0.25% triton X-100 in PBS. Slides were incubated in antibody for 24–48 hours and subsequently washed once for 30 minutes and twice for 1 hour in PBS. Secondary antibody was prepared in 2% BSA in PBS. Slides were incubated 24 hours, subsequently washed once 30 minutes and twice for 1 hour and mounted with Vectashield (Vector Labs, CA, USA). Images were analyzed as described for PI staining. All details concerning antibodies are presented in S1 Table.

## Sholl analysis method for microglial activation

We used CX3CR1-GFP[+]/- mice pups to set up the slice culture to do Sholl analysis to quantify microglial projections. Unbiased Sholl analysis was done by selecting 3–4 representative microglial cells in each max projection image using Fiji (ImageJ). Each image was thresholded by eye to convert each 32-bit grayscale image to a compatible 8-bit image with an inverted LUT. The freehand selection tool was utilized to appropriately select the desired microglial cell and clear the outside. The straight-line tool was used to define the largest Sholl radius, beginning in the center of the cell and arbitrarily extending outward. Sholl analysis was performed

under the 'most informative' normalized profile, which auto-determines whether to use a semi-log or log-log method of analysis [27]. Area was used as the normalizer in each profile. Parameters were set with a beginning radius of 5 microns, an ending radius of 49 microns, and a step size of 2 microns. The enclosing radius cutoff was set at 1 intersection. Sholl analysis data was presented by averaging the number of intersections per step size per group.

## Isolation of human umbilical cord and adult peripheral blood mononuclear cells

Freshly collected human umbilical cord blood was provided by the Carolinas Cord Blood Bank at Duke, an FDA licensed public cord blood bank that accepts donations of cord blood collected after birth from the placentas of healthy term newborns after written informed consent from the baby's mother. Also with maternal informed consent, cord blood units not qualifying for banking for transplantation were designated for research and made available for this study. Peripheral blood (PB) was obtained via venipuncture from healthy adult volunteer donors. Procurement of human samples were obtained using protocols approved by the Duke University Institutional Review Board. Mononuclear cells were isolated from CB and PB by density centrifugation using standard Ficoll-Hypaque technique (GE Healthcare) then treated with 0.15M $NH_4Cl$ to lyse residual erythrocytes and washed in phosphate-buffered saline (PBS).

## Immunomagnetic cell isolation of various sub-populations of CB and PB experiments

Specific sub-populations were isolated or removed from CB-MNC or PB-MNC by immuno-magnetic sorting using EasySep cell kits for human $CD34^+$, $CD3^+$, $CD14^+$ and $CD19^+$ cells (Stemcell Technologies, Vancouver, Canada, Catalog #18096, #18051, #18058 and #18054 respectively) following the manufacturer's directions. Flow-through fractions from positive selection columns were re-run through the columns to increase the purity of targeted populations. A sample of each cell preparation was analyzed by flow cytometry to determine cellular composition [28]. Immuno-magnetically selected specific sub-population of cells with ≥80% purity was used for any experiment. More highly purified populations were obtained by cell sorting as described below for gene expression analysis.

## RNA isolation and microarray analysis

RNA isolation and microarray analysis were carried out exactly as described previously using 54,675 probe set Affymetrix GeneChip Human Transcriptome Array 2.0 microarrays and Partek Genomics Suite 6.6 (Partek Inc., St. Louis, MO) software for analysis [28]. S1 Table outlines the experimental methods to prepare cells used for RNA extraction, the number of donors, and the characteristics of the donors used for each chip. Full expression analysis of CB data from Experiment 714 was previously published [28]; comparison of CB to PB-$CD14^+$ cells was not included in that publication. S2 Table provides demographic information on the donors and describes the preparation of $CD14^+$ cells used for the analysis, and numbers we used to designate the experiments in the text.

## Western blotting

Western blotting was carried out as previously described [28] using antibodies described in S1 Table.

## Statistical analysis

Data analysis was performed by calculating the mean of the values for each individual group ± standard error of mean and shown, graphically. Statistical analyses were carried out with GraphPad Prism software. All comparisons were performed by one-way analysis of variance (ANOVA) followed by post-hoc analysis with Bonferroni correction. Mean differences were considered significant if $p < 0.05$ was computed.

## Results

### CB-MNC protect organotypic brain slice culture cells from OGD induced damage

Mouse brain organotypic slice cultures were exposed to OGD for 1-hour and returned to the normoxic condition with media containing glucose for the cell treatment. A schematic diagram of the organotypic brain slice culture system is shown in S1 Fig. The extent of damage in the brain slice culture was evaluated by cellular PI uptake after the ischemic insult and following cell treatment. We found a significant number of cells in mouse brain slice cultures became permeable to PI during the three days following OGD shock Fig 1A. Addition of 25,000 CB-MNC to the surface of each OGD-shocked brain slice reduced the number of PI-stained dead cells significantly compared to the OGD-shocked control slices cultured without CB cells. This decrease in the number of dead cells in OGD-shocked cultures protected by CB-MNC approached background cell death in normoxic cultures. Quantitative analysis of PI-positive cells showed that the protective effect of CB-MNC was dose dependent between 2,500 to 25,000 CB-MNC per slice; only 25,000 CB-MNC/slice gave statistically significant ($p < 0.01$) protection (Fig 1B). Accordingly, we used this dose of cells for all other subsequent experiments. When we added CSFE-labeled CB-MNC directly onto brain slices, we detected green fluorescent cells on the membrane near to and on the s 72 hours after the cell addition (S1 Fig). Staining with antibody to human nuclear antigen confirmed that these CFSE labeled cells were human cells (S1 Fig).

To determine if paracrine factors released from CB-MNC contribute to their neuroprotective effects on OGD-shocked brain slices, we added CB-MNC to the medium below the membrane instead of directly onto the OGD-shocked slices. This prevented direct contact between CB-MNC and brain cells, but permitted agents secreted by CB-MNC to access to the cells in the brain slices through the 0.4μm pores. To compensate for possible dilution of protective factors by the large volume the culture medium, we added $1.25 \times 10^5$ cells below the membrane. Adding CB-MNC below the membrane, directly into the medium, reduced brain cell death (Fig 1C). Thus, neuroprotection by CB-MNC after OGD is mediated at least in part through secreted factors.

To identify what cell-types within CB-MNC mediated neuroprotection following OGD, we tested the ability of CB-MNC depleted of specific cell populations as well as isolating specific populations of cell from CB-MNC in our above organotypic cell death assay. Immunomagnetically enriched CD14+ monocytes from CB were sufficient to protect slices from OGD, and CB-MNC depleted of CD14+ monocytes were no longer able to confer protection (Fig 1D). Depleting other populations, e.g CD3+ T-lymphocyte or CD19+ B-lymphocytes or CD34+ hematopoietic progenitor cells from CB-MNCs did not block protection.

### CB CD14+ monocytes protect neurons and reduce glial activation

CB-CD14+ monocytes preserved neurons and dampened microglial and astrocytic activation following OGD. Cell death following OGD was mirrored by a large decrease in the NeuN-

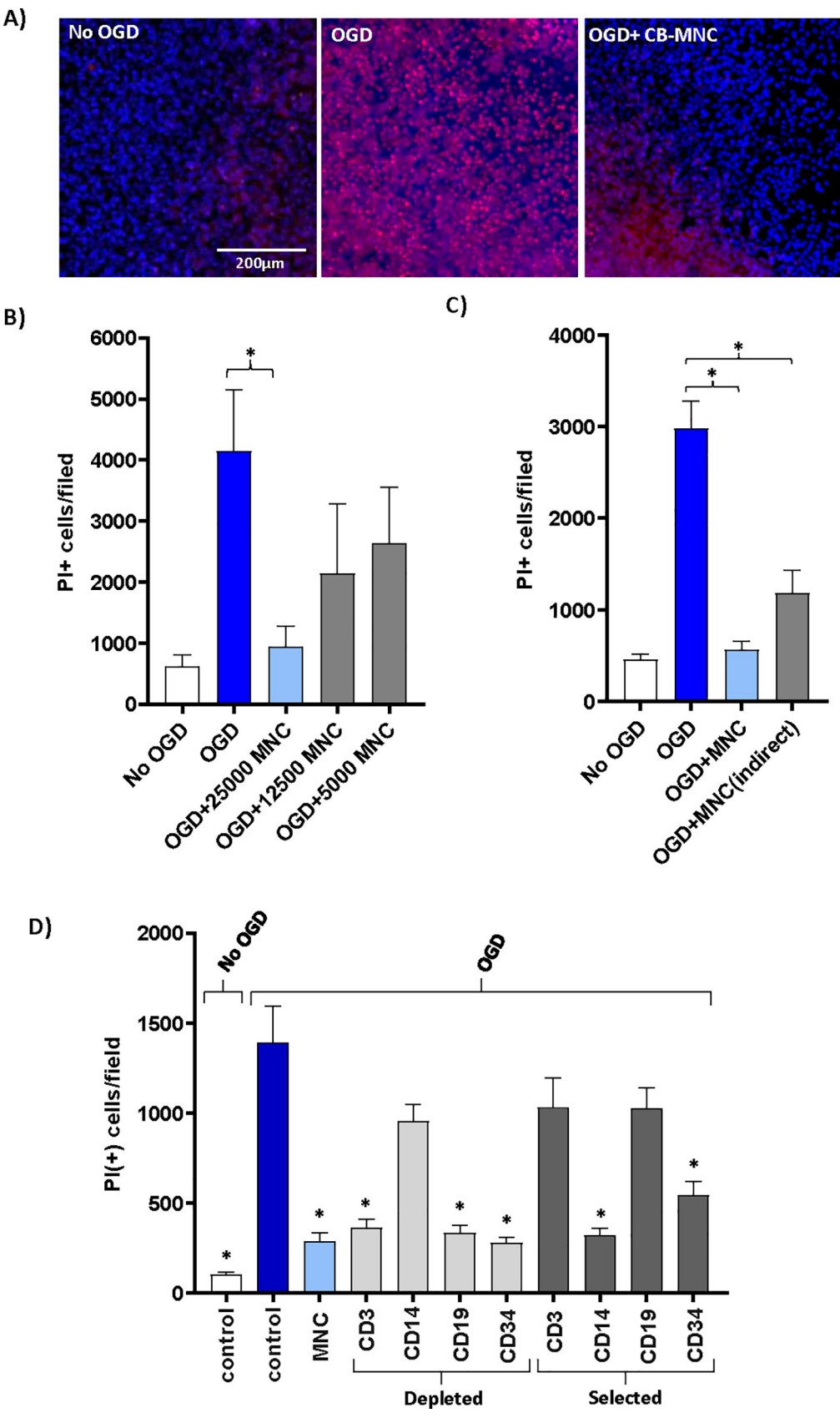

**Fig 1. Human cord blood mononuclear cells reduce death of mouse forebrain cells following OGD shock.** A) Left panel shows slice cultures not exposed to OGD. Slices in other panels were exposed to OGD for one hour, returned to normoxic, glucose replete conditions, and then cultured for 72 hours after which cell viability was assayed by staining with DAPI (blue) and PI (red). Middle panel shows slices cultured without added CB cells. Right panel shows slices cultured with 25,000 CB-MNC added directly onto slice at the end of OGD. B) Protection of brain cells following OGD depends on dose of CB-MNC added to slices. PI-stained cells were counted in contiguous 10X high power fields in the periventricular region. Bar graphs show mean $^+$/- SE of PI-stained cells per 10X high-power filed. Only the 25,000 cell dose group showed protection (n = -3, one way ANOVA, $^*$ p≤0.001). C) Paracrine factors from CB-MNC protect brain slice cultures after OGD shock. CB MNC were added either onto slice (light blue bar, $2.5\times10^4$ cells) or in medium below membrane (grey bar, $1.25\times10^5$ cells; n = 3, one way ANOVA, $^*$ p<0.01). D) OGD-shocked slices were co-cultured with CB-MNC that had been immunomagnetically depleted of the specific subpopulations or were co-cultured with immunomagnetically selected subpopulations expressing the surface antigen shown. First column on the left shows normoxic controls. All other data from OGD shocked slices (one-way ANOVA; $^*$p<0.001).

positive neuronal nuclei (Fig 2A and Fig 2B). Astrocytes in the cultures that were treated with OGD became hypertrophic and extended multiple processes taking on a characteristic acti-vated morphology (Fig 2C) [29, 30]. Astrocytes were less activated in CB-CD14$^+$ treated slices than CD14-depleted CB-MNC or untreated OGD slices (Fig 2C). It is well established that microglial change morphology from a highly ramified resting state to a reactive/amoeboid state upon ischemic insult [31]. To visualize the microglial morphological change much easily we initiated the organotypic brain slice cultures using CX3CR1-GFP$^+$/- mice P2 pups. Sholl analysis of these morphological changes reflecting microglial activation showed that CB-monocytes prevented microglial proliferation and activation in slices exposed to OGD (Fig 2D–Fig 2F).

## PB-MNC or purified PB-CD14$^+$ cells failed to protect from OGD induced tissue damage

Since PB-MNC are plentiful in patients with unresolved HI induced injuries we hypothesized that PB-MNCs would not protect against OGD insult. Unlike CB-MNCs, PB-MNCs, CD14$^+$ depleted PB-MNC, or isolated CD14$^+$-PB monocytes were unable to prevent cell death (Fig 3A) or loss of neurons (Fig 3B), following OGD insult.

The differences in activity between CB-MNC and PB-MNC provided an approach to begin to explore the mechanisms by which CB-MNC protect brain cells from hypoxic injury. We reasoned that transcripts for mechanistically important factors would be over-expressed in CB-MNC relative to PB-MNC. To identify these transcripts, we compared whole transcrip-tome microarrays analysis of CB- and PB-MNC. As described in the Supporting Information section, in all, we analyzed seven adult PB donors and seven CB donors in two separate experi-ments (714 and 1213) and found that CB and PB-CD14$^+$ monocytes have unique mRNA expression profiles. A heat map presentation of the data analysis of experiment experiments 1213 (Fig 4A), for example, shows that CB and PB-CD14$^+$ monocytes differentially expressed 1553 transcripts. Of these, 474 probes detected transcripts expressed only in PB-CD14$^+$ mono-cytes, and another 204 probes detected transcript only expressed in CB-CD14$^+$ monocytes. CB and PB-CD14$^+$ monocytes fall into discrete populations defined by these differentially expressed transcripts.

Since CB monocytes protect at least in part through secreted factors, we determined which differentially expressed genes (identified in both experimental analyses) encoded secreted pro-teins or proteins that directly synthesized secreted products. Seven candidates emerged from this analysis (Table 1). We next analyzed these seven candidate proteins in cell lysates from CB and PB CD14$^+$ monocytes (Fig 4C). CB monocytes expressed more CHI3L1, INHBA, IL10, MMP9, and TSP1 than PB-CD14$^+$ monocytes. Cystathionine (CTH) and VEGFA were strongly, but not differentially, expressed by both CB- and PB-MNC (S3 Fig). We also used

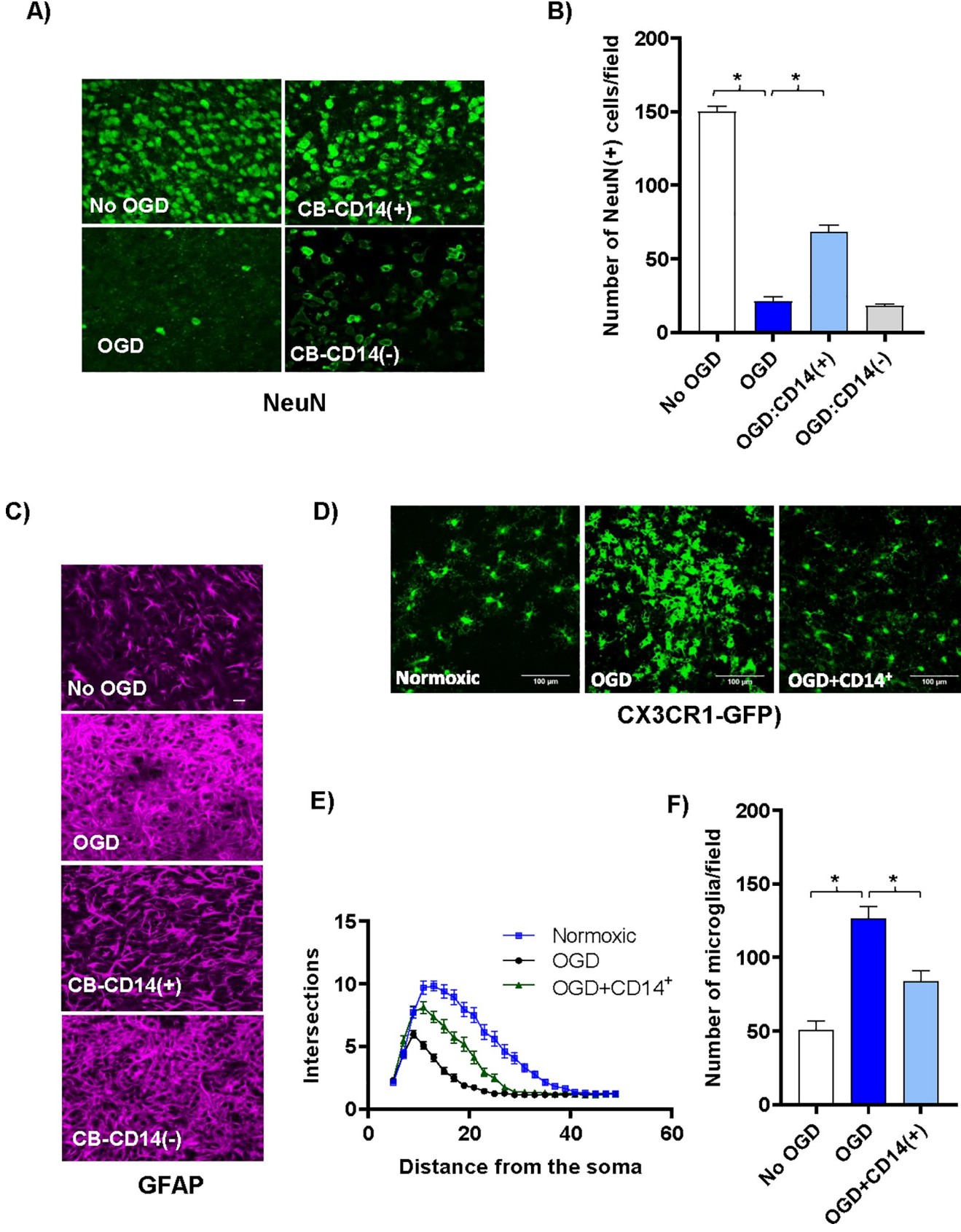

**Fig 2. CB CD14+ monocytes protect neurons and reduce glial activation following OGD shock.** (A) Confocal images (40x) of antibody stained neurons (green, anti-NeuN) in the periventricular region of control and cell treated brain slice cultures co-cultured three days with various CB cell populations following OGD shock. Top left panel shows control slices not exposed to OGD and cultured without human cells. All other panels show slices exposed to OGD prior to addition of human cells. Top right panel shows slices co-cultured with selected CB-CD14+ monocytes; and bottom right panel, with CB MNC depleted of CD14+ monocytes. (B) The average number of NeuN+ neurons in 40x high-powered fields (HPF) located along the periventricular region was determined. Values shown are means +/- standard deviation. N = 3 slices under each condition. Statistically significant differences (p<0.01) compared to the OGD control are indicated by asterisks. (C) Confocal images (40x) of antibody stained astrocytes [magenta, anti- GFAP] in the periventricular region of control and cell treated brain slice cultures co-cultured three days with various CB cell populations following OGD shock. Top row shows control slices not exposed to OGD and cultured without human cells. All other rows show slices exposed to OGD prior to addition of CB cells. Third row shows slices co-cultured with CB-CD14+ monocytes; fourth row, with CB MNC depleted of CD14+ monocytes. (D) Representative confocal images of microglial cells in CX3CR1-GFP+/- mouse brain slice cultures and numbers are shown. (E) Sholl profiles of microglial cells in brain slices of normoxic, OGD-shocked and OGD-shocked treated with CBCD14+ cells. Intersections were counted at 2μm intervals from the soma center to a radius of 5 μm to 50 μm. Curves represent mean intersection values ±SEM. (F) Number of microglia in control and OGD-treated brain slice cultures with and without added CB CD14+ cells.

immunocytochemistry to determine how CHI3L1, MMP9 and TSP1 proteins were expressed within CB and PB-CD14+ monocyte populations. S2 Fig shows that CHI3L1 and TSP1 were more strongly expressed in CB than PB-CD14+ monocytes and that these two proteins were present in virtually all CD14+ monocytes. CB monocytes also expressed more MMP9 than PB monocytes, but in this case, expression was confined to a subpopulation of CD14+ monocytes that was less common in PB monocyte populations.

## Discussion

We demonstrated that CB-MNC, specifically the CB-CD14+ cells, protect neurons from death after OGD insult. Depleting CD14+ cells, but not other cell types, abrogated the neuroprotective effects of CB-MNC. Purified CD34+ cells also have neuroprotective activity, but given that depleting CD34+ cells did not alter neuroprotection and that CD14+ cells are 10-50-fold more abundant in CB MNC than CD34+ cells, we attribute the neuroprotective activity of CB-MNC in our assay system to CD14+ cells. Neuroprotection was mediated primarily by soluble factors produced by CD14+ monocytes. This corroborates previous studies demonstrating that infiltrating monocytes sequestered in the brain meninges modulate brain inflammation and promote repair following HI injuries [32, 33]. CB CD14+ monocytes used as a therapeutic agent may have similar effects whether administered alone as a selected subpopulation or as a component present in the total CB-MNC. These results are consistent with findings of Womble *et al*. [13] showing that purified CB CD14+ monocytes, and none of the other human CB mononuclear cell sub populations tested, were neuroprotective in the rat middle cerebral artery occlusion [MCAO] model.

Unlike CB monocytes, PB monocytes had little or no impact on glial activation or cell death in the OGD assay. These difference in activities are consistent with the clinical observation that brain damage develops in patients following hypoxic injury even though peripheral blood cells, including peripheral blood monocytes circulate in very high numbers. Indeed, infiltration of peripheral blood monocytes is considered an important part of the pathogenesis of hypoxic brain injury. The same situation pertains to animal models. Bachstetter *et al*. reported that CB-MNC, but not PB-MNC, stimulated neurogenesis in aging rat brains [34]. CB, neonatal peripheral PB, and adult PB express different receptors, secrete different cytokines, and respond differently to inflammatory stimuli [35]. If these differences are reflected in the differential protective activity in the OGD assay remains to be determined.

We identified differentially expressed genes enriched in CB monocytes compared to PB monocytes. Based on our transwell experiments and other published data [10, 14] demonstrating that CB-MNC mediated repair of brain tissue through paracrine factors, we focused on finding secretory molecules over expressed in CB monocytes. Proteins encoded by five (*CHI3L1*, *TSP1*, *MMP9*, *IL10*, and *INHBA*) of the seven candidate genes we initially identified

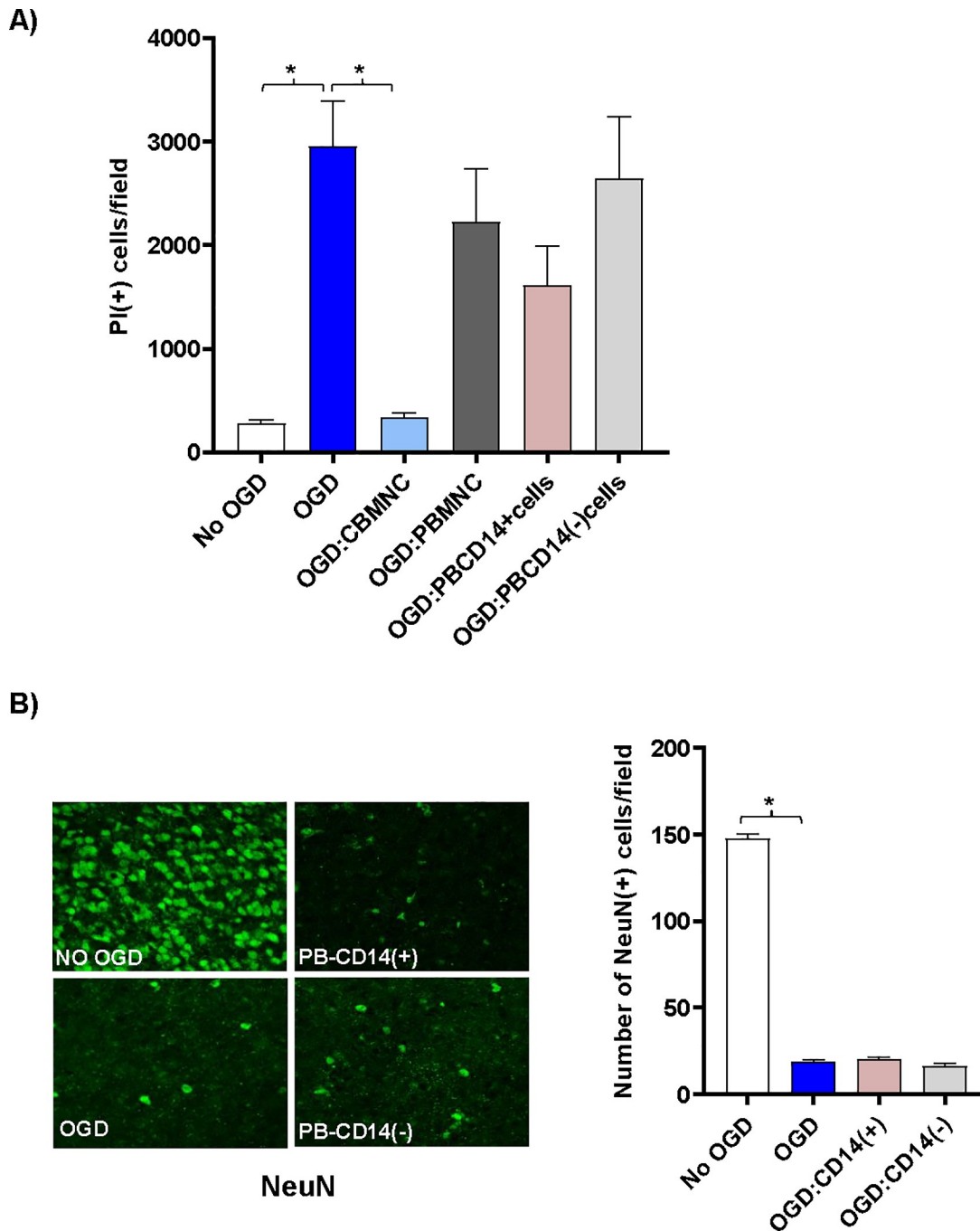

**Fig 3. CD14+ CB, but not PB, monocytes protect neurons following OGD shock.** (A) CB-MNC, PB-MNC, CD14+ or CD14 depleted PB cells (25,000cells/slice) were added directly onto OGD shocked brain slice cultures as indicated, and PI-positive cells dead were quantified and graphically represented. Statistically significant differences determined by one-way ANOVA (p<0.001) compared to the OGD control are indicated by asterisks. (B) Confocal images (40x) of antibody stained neurons [green, anti-NeuN] of control and cell treated brain slice cultures co-cultured three days with various PB cell populations following OGD shock. Top left panel shows control slices not exposed to OGD and cultured without human cells. All other panels show slices exposed to OGD prior to addition of human cells. Top right panel shows slices co-cultured with PB-CD14+ monocytes; bottom right panel, with PB MNC depleted of CD14+ monocytes. In the right panel bar-graph plot is shown, the average number of NeuN+ neurons, within sequential 40x high-powered fields located along the periventricular region was determined. Values shown are means +/- standard deviation. n = 3 slices under each condition. Statistically significant differences (p<0.01) compared to the OGD control are indicated by asterisks.

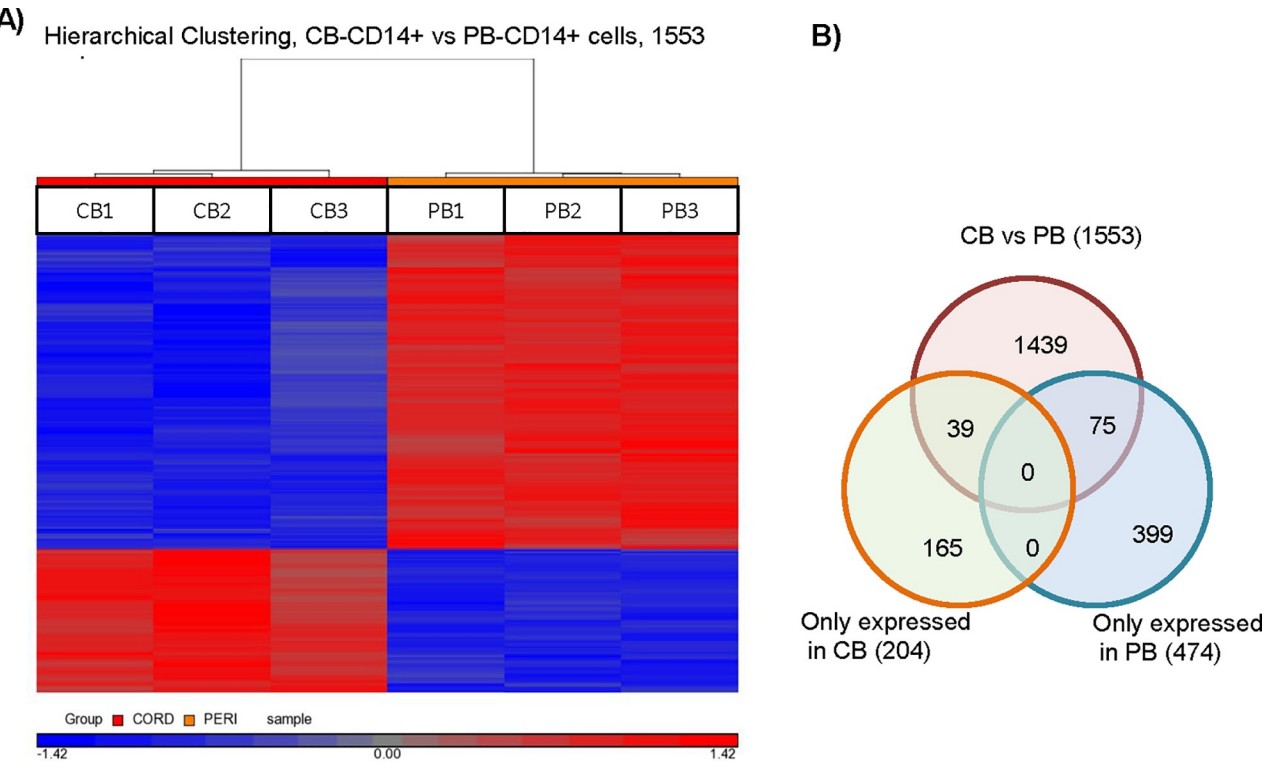

A) Hierarchical Clustering, CB-CD14+ vs PB-CD14+ cells, 1553

B) CB vs PB (1553)

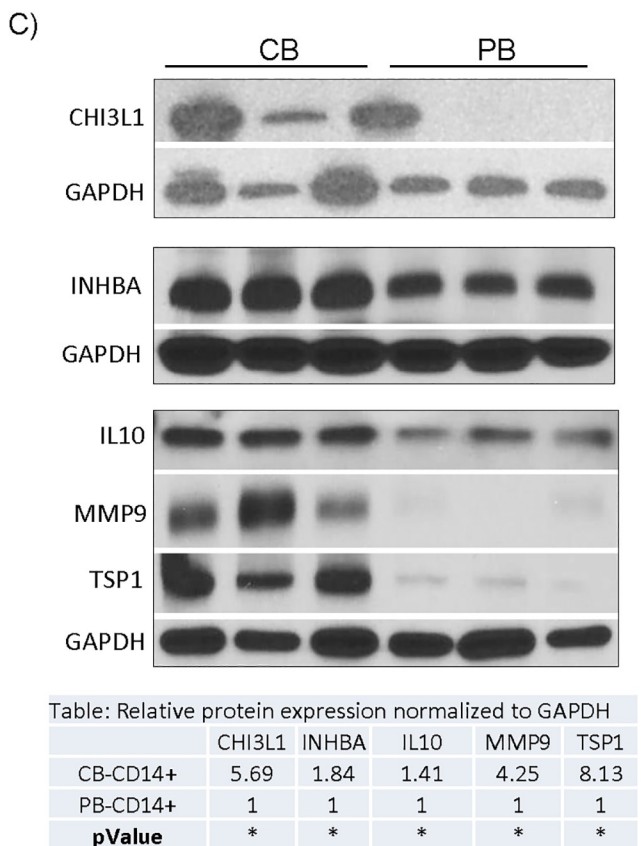

C)

Table: Relative protein expression normalized to GAPDH

|  | CHI3L1 | INHBA | IL10 | MMP9 | TSP1 |
|---|---|---|---|---|---|
| CB-CD14+ | 5.69 | 1.84 | 1.41 | 4.25 | 8.13 |
| PB-CD14+ | 1 | 1 | 1 | 1 | 1 |
| **pValue** | * | * | * | * | * |

**Fig 4. Identification of secretory proteins expressed by CB CD14[+] cells that may mediated protections against OGD.** (A) Comparative whole transcriptome analysis of CB-CD14[+] and PB-CD14[+] cells. Heat maps from Experiment 1213 showing differentially expressed probes in CB (CB-CD14[+]) and PB (PB-CD14[+]) cells. Up and downregulated genes are displayed in red and blue, respectively. RMA analysis detected significantly (p<0.05) different expression (at least two-fold) of probe sets corresponding to 1553 genes. (B) Gene expression comparisons between CB-CD14[+] and PB-CD14[+] cells by Venn diagram. Genes in overlapping sets show the differential expression in two or three comparison pairs. (C) Protein expression analysis of CB-CD14[+] and PB-CD14[+] cells. a) Lane 1–3, represent three different samples (n = 3) of CB-CD14[+] cells and Lane 4–6 represent, three different samples (n = 3) of PB-CD14[+] cells. The results confirmed enrichment of CH3L1, INHBA, IL-10, matrix metalloproteinase-9 (MMP9) and TSP1 in CB-CD14[+] relative to PB-CD14[+] monocyte homogenates. GAPDH was used as loading control. Quantitative expression of each proteins is shown in the table. Statistical significance (p< 0.05) is shown by asterisks.

were more abundant in homogenates of CB than PB monocytes. TSP1 [36–40], CHI3l1 [41–44]; MMP9 [45–49], IL10 [50], and INHBA [51, 52] can all promote tissue repair, including repair in the brain. TSP1, CHI3l1 and MMP9 showed the largest difference in protein expression, and CB monocytes have more of these three proteins in cytoplasmic granules, presumably secretory granules, than PB monocytes. Thus, CHI3l1, TSP1, and MMP9 may be particularly important in paracrine mechanisms by which CB monocytes reduce glial activation and protect brain neurons from OGD. Furthermore, correlating the biological and clinical activities with expression of these markers may provide a path to a biologically based potency assay for CB products in brain repair indications.

Though our work suggests that secretory proteins CHI3l1, TSP1, and MMP9 from monocytes might contribute to neuroprotection, other important protective gene products are probably induced by CB monocytes in or near HI-shocked brain tissue. The OGD-shocked brain slice model should be useful in identifying these gene products and elucidating more precisely how CB monocytes intervene in the pathogenic process.

**Table 1. Seven candidate genes encoding secreted factors over-expressed by CB compared to PB-CD14[+] monocytes.** All probes sets detecting each candidate genes in both microarrays are shown. Cord and peripheral blood donors are described in S1 Table. See text for screen used to identify candidates. P values are derived from RMAD analysis. Notes show MS5 analysis and indicate whether transcripts were detected exclusively in CB [CB only] or in both CB and PB-CD14[+] cells [CB>PB].

| Gene Symbol | Gene Title | Probe set | Chip Experiment 1213 (n = 4) | | | Chip Experiment 714 (n = 3) | | |
|---|---|---|---|---|---|---|---|---|
| | | | p-value | Fold-difference | Note | p-value | Fold-difference | Note |
| CTH | cystathionase | 217127 _at | 5.70E-04 | 41.9 | CB only | 1.18E-08 | 105.1 | CB>PB |
| | | 206085_s_at | 3.44E-03 | 14.3 | CB only | 8.14E-08 | 26.0 | CB only |
| CHI3L1 | chitinase 3-like 1 | 209395_at | l.33E-02 | 12.6 | CB only | 4.99E-02 | 5.0 | CB only |
| | | 209396_s_at | 1.83E-02 | 10.6 | CB only | 1.75E-01 | 2.8 | CB = PB |
| THBS1 | thrombospondin 1 | 215775_at | 6.59E-03 | 3.1 | CB>PB | 4.71E-03 | 2.6 | CB>PB |
| | | 201107 _s_at | 1.74E-02 | 3.5 | CB>PB | 5.84E-04 | 3.7 | CB only |
| | | 201109_s_at | 1.19E-04 | 32.1 | CB>PB | 5.56E-02 | 9.3 | CB>PB |
| | | 20 1108_s_at | 7.37E-04 | 20.5 | CB only | 8.98E-03 | 8.4 | CB>PB |
| | | 201110_s_at | 3.78E-05 | 22.2 | CB>PB | 1.98E-Ol | 4.3 | CB>PB |
| | | 235086_at | 2.88E-04 | 35.0 | CB>PB | 1.78E-02 | 8.4 | CB > PB |
| | | 239336_at | 2.24E-03 | 8.8 | CB only | 4.42E-03 | 5.5 | CB only |
| MMP9 | matrix metallopeptidase 9 | 203936_s_at | 1.18E-03 | 13.8 | CB>PB | 1.17E-02 | 5.4 | CB>PB |
| IL10 | interleukin 10 | 207433_at | 1.51E-02 | 2.5 | CB>PB | 3.44E-03 | 2.8 | CB>PB |
| VEGF-A | vascular endothelial growth factor -A | 210512_s_at | 1.18E-03 | 3.6 | CB>PB | 1.91E-Ol | 2.0 | CB = PB |
| | | 212171_x_at | 2.77E-04 | 4.4 | CB>PB | 3.38E-02 | 2.2 | CB>PB |
| | | 210513_s_at | 2.09E-03 | 4.1 | CB>PB | 4.97E-02 | 1.9 | CB = PB |
| | | 211527 _x_at | 1.35E-03 | 5.1 | CB>PB | 5.31E-02 | 2.9 | CB > PB |
| INHBA | inhibin, beta A | 227140_at | 6.07E-03 | 10.5 | CB only | 4.36E-02 | 14.3 | CB>PB |
| | | 210511_s_at | 2.28E-02 | 2.3 | CB>PB | 6.75E-02 | 3.9 | CB>PB |
| | | 204926_at | not detected | | | not detected | | |

Finally, we note that although our standard brain slice model preserves many important aspects of brain architecture and neuron-glial interactions in response to OGD stress, the system presented here does not replicate all important aspects influencing cell therapy for HI-brain injury. We have not yet explored whether CB CD14+ monocytes can reverse neural death when added to cultures at longer periods after shock or whether slice cultures from adult brain slices will be protected as efficiently those from neonates. Addressing these issues should be straightforward in this system. Also, the slice culture system in which candidate cell therapy populations are added directly to brain slices, or in a small amount of medium directly below the slices, may not reflect the dosing, biodistribution, or pharmacokinetics associated with any of the routes (intravenous, intrathecal, intra-arterial, intraparenchymal) that have been used to administer CB-MNC and other cell therapies to experimental animals or patients with HI-induced brain injury. How each of these routes impacts dosing or targeting of cells to the brain is not yet clear, even after intravenous injection, the most common route of administration. Animal studies have shown that some unidentified CB cells are present near brain lesions for short periods of time following intravenous treatment of acute stroke [53, 54] or neonatal HIE with CB-MNC [55] but the function of these cells is unclear. Some evidence suggests that CB-MNC respond to chemokines by migrating to ischemic brain regions [56–58]. Womble et al. found that the beneficial activity of intravenously injected CB-MNC in a rat stroke model resided in the CD14$^+$ monocyte population. How many CB-MNC or monocytes that reach the brain following intravenous injection in patients with HI-induced brain injury is unknown. Indeed, some animal studies have suggested that intravenously injected CB-MNC products [59] do not need to reach the brain in order to promote repair of stroke or other HI brain injury. Instead, cell products reaching the lungs or spleen may induce endogenous cells to produce soluble factors or activated cells that go to the brain and mediate repair [60–62]. As already noted, our results demonstrating the neuroprotective activity of CB CD14+ correlate strongly with results in the MCAO rat model, and this suggests that studies investigating the biodistribution of CD14+ CB monocytes following different routes in the MCAO model and in neonatal and adult rodent hypoxia models maybe useful in elucidating where interactions important to therapeutic outcome occur. In summary, monocytes in CB, but not PB, protect brain neurons from death and reduce glial activation following HI insult in an *in vitro* OGD model. Soluble factors released from CB monocytes contribute to this protection. We have identified secreted proteins enriched in CB CD14$^+$ monocytes compared to PB monocytes that may play a role in neuroprotection and repair. This work enables future detailed study of the mechanism of neuroprotection and development of mechanism-based release assays for CB products, and formulation of new strategies for using CB monocytes as therapeutic agents in treatment of HI-induced brain injuries.

## Supporting information

**S1 Fig.**
(TIF)

**S2 Fig.**
(TIF)

**S3 Fig.**
(TIF)

**S1 File.**
(DOCX)

**S1 Table.**
(DOCX)

**S2 Table.**
(DOCX)

**S3 Table.**
(DOCX)

## Acknowledgments

The authors are grateful to Susan Buntz for helping with flow-cytometry, the staff at the Carolinas Cord Blood Bank for providing cord blood units, to Dr. Michael Cook at the Duke Cancer Center Flow Cytometry Facility for sorting cells, and to Zhengzheng Wei at the Duke Institute for Genomic Sciences Microarray Core Facility for performing microarray analyses. This work was supported by grants from the Julian Robertson Foundation and the Marcus Foundation.

The authors also wish to acknowledge the friendship and the daily advice on both practical and theoretical matters given freely to them by Dr. Bob Storms during the course of this research. Bob died on May 22, 2017.

## Author Contributions

**Conceptualization:** Arjun Saha, Joanne Kurtzberg, Andrew E. Balber.

**Data curation:** Arjun Saha, Sachit Patel, Li Xu, Paula Scotland, Jonathan Schwartzman.

**Formal analysis:** Arjun Saha, Sachit Patel, Anthony J. Filiano, Andrew E. Balber.

**Funding acquisition:** Joanne Kurtzberg, Andrew E. Balber.

**Investigation:** Arjun Saha, Joanne Kurtzberg, Andrew E. Balber.

**Methodology:** Arjun Saha.

**Project administration:** Arjun Saha, Andrew E. Balber.

**Resources:** Arjun Saha, Joanne Kurtzberg, Andrew E. Balber.

**Supervision:** Arjun Saha, Joanne Kurtzberg.

**Validation:** Arjun Saha, Andrew E. Balber.

**Visualization:** Arjun Saha.

**Writing – original draft:** Arjun Saha, Andrew E. Balber.

**Writing – review & editing:** Arjun Saha, Anthony J. Filiano, Joanne Kurtzberg, Andrew E. Balber.

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
