## [Decision Letter · Decision Letter 0]

11 Jul 2019

PONE-D-19-16233

Human Umbilical Cord blood monocytes, but not adult blood monocytes, rescue brain cells from hypoxic-ischemic injury: Mechanistic and therapeutic implications

PLOS ONE

Dear Dr Saha,

Thank you for submitting your manuscript to PLOS ONE. After careful consideration, we feel that it has merit but does not fully meet PLOS ONE’s publication criteria as it currently stands. Therefore, we invite you to submit a revised version of the manuscript that addresses the points raised during the review process.

We would appreciate receiving your revised manuscript by Aug 25 2019 11:59PM. To enhance the reproducibility of your results, we recommend that if applicable you deposit your laboratory protocols in protocols.io, where a protocol can be assigned its own identifier (DOI) such that it can be cited independently in the future. For instructions see: http://journals.plos.org/plosone/s/submission-guidelines#loc-laboratory-protocols

We look forward to receiving your revised manuscript.

Kind regards,

Cesar V Borlongan

Academic Editor

PLOS ONE

Journal Requirements:

2. To comply with PLOS ONE submissions requirements, please provide methods of sacrifice in the Methods section of your manuscript. Additionally, please provide additional details regarding participant consent for providing blood samples. In the ethics statement in the Methods and online submission information, please ensure that you have specified (1) whether consent was informed and (2) what type you obtained (for instance, written or verbal, and if verbal, how it was documented and witnessed). If the need for consent was waived by the ethics committee, please include this information.

3. Thank you for your ethics statement : "All experiments were performed in accordance with Duke University Institutional Animal Care and Use Committee’s policies and followed approved protocols."

Please amend your current ethics statement to confirm that your named ethics committee/IACUC  specifically approved this study.

For additional information about PLOS ONE submissions requirements for animal ethics, please refer to http://journals.plos.org/plosone/s/submission-guidelines#loc-animal-research  

Additional Editor Comments (if provided):

Both reviewers are highly enthusiastic of this paper and only suggested very minor revisions, which can be easily addressed by the authors.

Reviewers' comments:

Reviewer's Responses to Questions

**Comments to the Author**

1. Is the manuscript technically sound, and do the data support the conclusions?

Reviewer #1: Yes

Reviewer #2: Yes

2. Has the statistical analysis been performed appropriately and rigorously? 

Reviewer #1: Yes

Reviewer #2: Yes

3. Have the authors made all data underlying the findings in their manuscript fully available?

Reviewer #1: Yes

Reviewer #2: Yes

4. Is the manuscript presented in an intelligible fashion and written in standard English?

Reviewer #1: Yes

Reviewer #2: Yes

5. Review Comments to the Author

Reviewer #1: Saha et al used mouse brain organotypic slice cultures that were oxygen and glucose deprived (OGD) and then treated them with cord blood (CB) or adult peripheral blood (PB). They found that CB CD14+ mononuclear cells (MNCs) protected neurons from OGD-induced death and reduced both microglial and astrocyte activation which was not shown with PB mononuclear cells. The authors showed that the protective effect of the CB MNCs was mediated by secreted factors and did not require cell-to-cell contact with the injured brain.

1. This very interesting data all revolves around the OGD assay. Are there any other in vitro or even in vivo animal assays that could validate this system?

2. Were dose-response experiments performed with the peripheral blood mononuclear cells (MNCs) to insure that an adequate dose was tested?

3. Were experiments with peripheral blood MNCs added directly onto the slice vs into the medium below performed as they were for the CB MNCs?

4. Also were dose response experiments performed with the CB MNCs that were added directly to the medium?

5. The authors discuss the fact that the commonly employed and most feasible intravenous use of CB may have limitations when trying to correlate results with the direct application OGD assay. Do they believe there is a role for direct intravascular administration of CB into the injured brain?

6. Are there other models as discussed ni #1 above where these CB CD14+ MNCs could be infused IV to more accurately reflect the clinical situation?

Reviewer #2: Using in vitro mouse brain organotypic slice cultures after oxygen and glucose deprivation (OGD) as a model for hypoxic-ischemic (HI) brain injuries, the authors have investigated the potential of cord blood versus peripheral blood MNC co-culture to protect neurons from OGD-induced death as well as reduce microglial and astrocyte activation. The authors report several novel findings: cord blood is effective at protection while peripheral blood is not; protection seems to be facilitated by secreted factors as direct contact is not required; and the protective effect is associated with CD14+ monocyte fraction. Differential gene expression studies have identified several candidate secreted factors preferentially produced by cord blood monocytes, future experiments may begin to evaluate whether those factors (alone or in combinations) could substitute for the cell-based co-culture protective responses.

The authors should provide additional information regarding the following questions and/or discuss why these points are not directly relevant:

1. Have the authors evaluated peripheral blood CD14+ MNC obtained after G-CSF stimulation/mobilization in their in vitro neuroprotection model (as the monocyte/dendritic cell populations are very different as compared with non-mobilized PBMC).

2. The authors are using murine slice cultures obtained from very young 2-day old pups, which may have unique brain responses that are more “embryonic” in nature. Have similar experiments been attempted with samples from slightly older murine brains, to determine if equivalent neuroprotective activity can be observed in this perhaps more clinically-relevant situation?

3. Along the same lines, the current model exposes the sliced cultures to MNC therapy immediately after OGD treatment. Have experiments been conducted to evaluate neuroprotection if MNC therapy is delayed for several hours post-OGD?

4. Were any dose-response experiments above the 25,000 cell level per slice culture performed with the PB-MNC or other cell fractions to assess whether the observed differences in neuroprotective activity could be accounted for by a below threshold phenomena (similar to the CB-MNC dose-response data shown in Figure 1B)? Similarly, were dose-response experiments (above and below 125,000) performed in the below the membrane culture experiments to better assess the indirect secretion neuroprotective activity?

6. PLOS authors have the option to publish the peer review history of their article (what does this mean?). If published, this will include your full peer review and any attached files.

Reviewer #1: No

Reviewer #2: No

---

## [Author Response · Author response to Decision Letter 0]

8 Aug 2019

Dr. Cesar V Borlongan

Academic Editor

PLOS ONE

Dear Dr. Borlogan:

Thank you for arranging for the review of our manuscript PONE-D-19-16233 and for forwarding the reviewer’s comments to us. With this letter, we are submitting a revised version addressing their comments and, hopefully, positioning the manuscript for publication. We address each of the comments in detail below; our responses are presented in blue font. We appreciate the reviewers’ comments and suggestions and feel that the manuscript is better because we have addressed them in this revision. We hope that you will now find the manuscript acceptable for publication in PLOS ONE. 

Sincerely,

Arjun Saha, Ph.D.

Project Leader

Marcus Centre for Cellular Cures

Duke University School of Medicine

701 West Main Street

Chesterfield Building, Room 5413

Durham, NC 27701

USA

Ph: 919-684-3934

Fax:919-681-9760

Journal Requirements:

Comments: We followed the journal style while revising the manuscript.

2. To comply with PLOS ONE submissions requirements, please provide methods of sacrifice in the Methods section of your manuscript. Additionally, please provide additional details regarding participant consent for providing blood samples. In the ethics statement in the Methods and online submission information, please ensure that you have specified (1) whether consent was informed and (2) what type you obtained (for instance, written or verbal, and if verbal, how it was documented and witnessed). If the need for consent was waived by the ethics committee, please include this information.

Comments: We have added the method of sacrifice in Materials and Methods section, p. 7, line 14.

3. Thank you for your ethics statement : "All experiments were performed in accordance with Duke University Institutional Animal Care and Use Committee’s policies and followed approved protocols."

Please amend your current ethics statement to confirm that your named ethics committee/IACUC specifically approved this study.

Comments: We have amended this in p. 7, line 4-6 of the revised manuscript.

For additional information about PLOS ONE submissions requirements for animal ethics, please refer to http://journals.plos.org/plosone/s/submission-guidelines#loc-animal-research

Comments: We have added this data in a new supporting figure, S3 Fig, and accordingly changed the main text (p. 15, line 10) and revised supporting info, p. 3, line 6-11.

Additional Editor Comments (if provided):

Both reviewers are highly enthusiastic of this paper and only suggested very minor revisions, which can be easily addressed by the authors.

Reviewers' comments:

Reviewer's Responses to Questions

Comments to the Author

1. Is the manuscript technically sound, and do the data support the conclusions?

Reviewer #1: Yes

Reviewer #2: Yes

2. Has the statistical analysis been performed appropriately and rigorously? 

Reviewer #1: Yes

Reviewer #2: Yes

3. Have the authors made all data underlying the findings in their manuscript fully available?

Reviewer #1: Yes

Reviewer #2: Yes

4. Is the manuscript presented in an intelligible fashion and written in standard English?

Reviewer #1: Yes

Reviewer #2: Yes

5. Review Comments to the Author

Reviewer #1: Saha et al used mouse brain organotypic slice cultures that were oxygen and glucose deprived (OGD) and then treated them with cord blood (CB) or adult peripheral blood (PB). They found that CB CD14+ mononuclear cells (MNCs) protected neurons from OGD-induced death and reduced both microglial and astrocyte activation which was not shown with PB mononuclear cells. The authors showed that the protective effect of the CB MNCs was mediated by secreted factors and did not require cell-to-cell contact with the injured brain.

1. This very interesting data all revolves around the OGD assay. Are there any other in vitro or even in vivo animal assays that could validate this system?

Response: We are glad to know that the reviewer found our work interesting. As reviewed in the introduction and discussion of the manuscript, CB mononuclear cells have been tested in several in experimental systems of hypoxic injury, including OGD-induced injury, in animals and in culture, and also used in related clinical trials. We decided to explore the OGD-brain slice system further because there is no consensus on the type of cells within CB mononuclear cells that mediate neuroprotection. We believe that the most important results presented in our manuscript are (1) that under defined experimental conditions, cord blood monocytes uniquely protect neurons and dampen glial activation in the brain slices exposed to OGD and (2) that peripheral blood monocytes do not. 

The reviewer asks how these results compare with results in other experimental systems. In response, we note that other CB cell types have been implicated in regulating neural and glial cell activities in cell culture systems as reviewed in the manuscript. In one well established animal model, cerebral artery occlusion in rats, the neuroprotective activity of intravenously administered CB mononuclear cells has been shown to reside in the CD14+ cell population [Reference #13 in the manuscript; Womble TA et al., 2014]. Furthermore, another study demonstrated that rat peripheral blood mononuclear cells alone had no effect in the MCAO model [Wu et al., Cell Transplantation, Vol. 26, pp. 571–583, 2017]. Thus, our results are consistent with these animal results. We have cited these references in the revision and emphasized the corroboration [see p. 16, line 18-21 (…These results are consistent with findings of Womble et al. [13] showing that purified CB CD14+ monocytes, and none of the other human CB mononuclear cell sub populations tested, were neuroprotective in the rat middle cerebral artery occlusion [MCAO] model….)]. 

Beyond the experimental systems, these initial results are also consistent with the clinical setting. Brain damage develops in patients following hypoxic injury even though peripheral blood cells, including peripheral blood monocytes circulate in very high numbers. Indeed, infiltration of peripheral blood monocytes is considered an important part of the pathogenesis of hypoxic brain injury. We note that the same situation pertains to animal models. In the clinic, infusion of cord blood mononuclear cells (usually including 5-10% monocytes) has been found to be safe and trials of potential therapeutic efficacy, based on observations in animal models, are continuing. We have added sentences to the discussion to emphasize this point [see p. 16, line 18-21 and p. 19, line14-18(….As already noted, our results demonstrating the neuroprotective activity of CB CD14+ correlate strongly with results in the MCAO rat model, and this suggests that studies investigating the biodistribution of CD14+ CB monocytes following different routes in the MCAO model and in neonatal and adult rodent hypoxia models maybe useful in elucidating where interactions important to therapeutic outcome occur..)

Thus, our results are consistent with other clinical and experimental observations. Further studies using CD14+ cord blood cells in rodent neonatal hypoxic-ischemic or adult middle cerebral artery occlusion (MCAO) models could be useful to exploring how closely the results in the brain slice system reproduce clinically relevant results. We have pointed this out in the penultimate paragraph of the revised discussion [p. 18, line 6-18 (….Finally, we note that although our standard brain slice model preserves many important aspects of brain architecture and neuron-glial interactions in response to OGD stress, the system presented here does not replicate all important aspects influencing cell therapy for HI-brain injury. We have not yet explored whether CB CD14+ monocytes can reverse neural death when added to cultures at longer periods after shock or whether slice cultures from adult brain slices will be protected as efficiently those from neonates. Addressing these issues should be straightforward in this system. Also, the slice culture system in which candidate cell therapy populations are added directly to brain slices, or in a small amount of medium directly below the slices, may not reflect the dosing, biodistribution, or pharmacokinetics associated with any of the routes (intravenous, intrathecal, intra-arterial, intraparenchymal) that have been used to administer CB-MNC and other cell therapies to experimental animals or patients with HI-induced brain injury. How each of these routes impacts dosing or targeting of cells to the brain is not yet clear, even after intravenous injection, the most common route of administration) and p. 19, line 14-18 (….As already noted, our results demonstrating the neuroprotective activity of CB CD14+ correlate strongly with results in the MCAO rat model, and this suggests that studies investigating the biodistribution of CD14+ CB monocytes following different routes in the MCAO model and in neonatal and adult rodent hypoxia models maybe useful in elucidating where interactions important to therapeutic outcome occur.)]. Also we edited the introduction section slightly in the revised manuscript [p. 6, line 4-7(….Organotypic slice cultures offer the advantage of preserving the cytoarchitecture of the tissue of origin and connectivity of different anatomical regions, as well as functional relationships and interactions between neighboring cells, such as neurons and astrocytes, keeping the intrinsic synaptic connections found in vivo)] to emphasize the advantage of using organotypic slice cultures over other in vitro systems.

2. Were dose-response experiments performed with the peripheral blood mononuclear cells (MNCs) to insure that an adequate dose was tested?

Response: At 25,000 cells/slice dose, CB-MNC, were significantly neuroprotective, but PB-MNC were inactive. As we found that monocyte cells are the active components of CB-MNC and we also know that average monocyte frequencies in cord blood and peripheral blood are similar [Immunology. 2011 May; 133(1): 41–50] we only tried the same dose for PB-MNC as we used the dose for CB-MNC. When we used isolated monocytes from CB and PB, both at a dose of 25,000cells/slice, in our OGD assay we could not find protective activity from PB-CD14+ monocytes, whereas CB-CD14+ monocytes were significantly effective (fig 3 and 2). And this is one of the major findings of our study presented here; at the same dose CB-CD14+ monocytes are neuroprotective but PB-CD14+ monocytes are not. Since, 25,000 PB-MNCs were not neuroprotective, we did not titrate cell number in the assay system as we did for CB-MNC. Also, please see our response to question #1.

3. Were experiments with peripheral blood MNCs added directly onto the slice vs into the medium below performed as they were for the CB MNCs?

Response: Since 25,000 PB-MNC did not significantly protect neurons when added directly onto the brain slices, we did not test the effect of PB-MNC added indirectly into the medium as the same paracrine effect would still be possible under the directly added cells on the tissue.

4. Also were dose response experiments performed with the CB MNCs that were added directly to the medium?

Response: We thank the reviewer for bringing up this point and edited the method section to make our methods more clear [P. 7, line 22-23 (….to check possible paracrine effect…) and P. 8, line 1-2 (…And in this case, to compensate for possible dilution of protective factors by the large volume the culture medium, we added 1.25x105 cells below the membrane…)] and results section [p. 13, line 9 (…directly into the medium)].

In principle, both cell to cell contact and paracrine effects could play a role in neuroprotection. Our goal in this experiment was simply to determine if CB CD14+ cells released neuroprotective soluble factors. We routinely found significant neuroprotection at 25,000 CB-MNC/slice and used this as a standard condition for direct addition of cells to slice cultures. When we decided to determine if we could detect paracrine effects, we decided to add 125,000 cells, 5-times more cells than were added directly to the slices, to try to account for the very large dilution of secreted factors in the culture medium [2µL of medium (containing the cells) on top of slice versus 1mL of medium below slice]. Our results show that CB monocytes do secrete protective paracrine factors, which were significantly neuroprotective at 125,000 cells/well dose, and we did not try any other doses. 

5. The authors discuss the fact that the commonly employed and most feasible intravenous use of CB may have limitations when trying to correlate results with the direct application OGD assay. Do they believe there is a role for direct intravascular administration of CB into the injured brain?

Response: We thank the reviewer for raising this very important point. We have modified the penultimate paragraph of the discussion [p. 18, line 6-18 and p. 19, line 14-18 (please see response to question #1)] to address these issues more fully. The best route for administering cell therapies to treat brain injury is a field of active investigation. Clinical trials using intravenous, intrathecal, intraarterial, intraparenchymal administration have been completed and are on-going. Comparing these trials is very complex as different agents have been used with different doses and devices. Another important issue is whether cells delivery to the brain is necessary to effect repair. Thus, delivery to other tissue may induce changes in those tissues that result in production of soluble proteins or cell populations that go to the brain to mediate repair. In one example possibly closely related to our work, the neuroprotective effect of CB mononuclear cells in the rat MCAO model was abrogated when animals were splectomized prior to injury, suggesting that the treatment effects depended on activities in the spleen. References are cited in the new paragraph. Further to our response to question 1, our results suggest that tracking biodistribution of CB CD14+ monocytes after infusion into animal models by different routes and how this correlates with neuroprotection may illuminate this issue further. All of these points are incorporated into the revision. 

6. Are there other models as discussed ni #1 above where these CB CD14+ MNCs could be infused IV to more accurately reflect the clinical situation?

Response: Please see response to questions #1 and #5. In addition, we note that the OGD-brain slice model that we have now standardized allows us to approach possible mechanisms of action and then to see how these may correspond to animal models and clinical outcomes. For example, we already have candidate secretory molecules made by CD14+ monocytes that could mediate neuroprotection directly or by regulating other cells. The culture system lets us explore this directly, and we can then look for the activities of these molecules in animal models. Soluble molecules can also be measure in patient samples. 

Reviewer #2: Using in vitro mouse brain organotypic slice cultures after oxygen and glucose deprivation (OGD) as a model for hypoxic-ischemic (HI) brain injuries, the authors have investigated the potential of cord blood versus peripheral blood MNC co-culture to protect neurons from OGD-induced death as well as reduce microglial and astrocyte activation. The authors report several novel findings: cord blood is effective at protection while peripheral blood is not; protection seems to be facilitated by secreted factors as direct contact is not required; and the protective effect is associated with CD14+ monocyte fraction. Differential gene expression studies have identified several candidate secreted factors preferentially produced by cord blood monocytes, future experiments may begin to evaluate whether those factors (alone or in combinations) could substitute for the cell-based co-culture protective responses.

The authors should provide additional information regarding the following questions and/or discuss why these points are not directly relevant:

1. Have the authors evaluated peripheral blood CD14+ MNC obtained after G-CSF stimulation/mobilization in their in vitro neuroprotection model (as the monocyte/dendritic cell populations are very different as compared with non-mobilized PBMC).

Response: This is an interesting point as G-CSF mobilized blood cells have been tested as potential stroke therapeutics. As we found monocytes are the bioactive cells in cord blood and we know that monocyte frequencies are similar in CB and PB, we have not explored this in our system and thank the reviewer for the suggestion.

2. The authors are using murine slice cultures obtained from very young 2-day old pups, which may have unique brain responses that are more “embryonic” in nature. Have similar experiments been attempted with samples from slightly older murine brains, to determine if equivalent neuroprotective activity can be observed in this perhaps more clinically-relevant situation?

Response: We understand the reviewer’s concern and as we mentioned in comment #1 of Reviewer-1, that other in vivo models could be used further to test our findings in a more clinically relevant situation but it is beyond the scope of our current proof-of-principle study. We used brain slices from young 2-day old mouse pups as these gives more consistent and reliable organotypic cultures compared to the brain slices from adult mice (Trends Neurosci. 1997 Oct;20(10):471-7). However, we would like to note (as mentioned in the methods section of the manuscript) that these slices were in the culture for about two weeks before we used those for OGD experiments. Here we wanted first know, as a proof of principle, using this ex vivo brain slice culture system, that whether CB-MNC and its various specific subpopulations of cells could be protective after hypoxic-ischemic shock. In response to this comment we have added how the age of the brain cells in the slice culture impacts response to CD14+ monocytes to a list of issues that can be resolved using the system described in the manuscript in order to further explore the clinical relevance; this appears in the new penultimate paragraph [p. 18, line 6-18 and p. 19, line 14-18 (please see response to question #1)]. 

3. Along the same lines, the current model exposes the sliced cultures to MNC therapy immediately after OGD treatment. Have experiments been conducted to evaluate neuroprotection if MNC therapy is delayed for several hours post-OGD?

Response: This is an important point raised by the reviewer. We have not studied the time course of the treatment effect further. As we mentioned earlier, the main focus of our study described here was to create the most favorable condition for the intervention so that we could see any beneficial effect and we believed that treating immediately after the insult might augment the probability of seeing protective effect. Also neuronal death in our system was very quick, we found significant neuronal deaths by 72-hours after the OGD shock, so we decided to treat these cultures as quickly as possible. Again, we have incorporated this point in the penultimate paragraph [p. 18, line 6-18 and p. 19, line 14-18 (please see response to question #1)] discussing important parameters that can be explored with the system.

4. Were any dose-response experiments above the 25,000 cell level per slice culture performed with the PB-MNC or other cell fractions to assess whether the observed differences in neuroprotective activity could be accounted for by a below threshold phenomena (similar to the CB-MNC dose-response data shown in Figure 1B)? Similarly, were dose-response experiments (above and below 125,000) performed in the below the membrane culture experiments to better assess the indirect secretion neuroprotective activity?

Response: Please see response #2, 3 and 4 of Reviewer-1. Furthermore, we found that CB-MNC at 25,000 cells/slice dose, were significantly neuroprotective, but PB-MNC were not. As we learnt that monocyte cells are the active components of CB-MNC and as it is also known that average monocyte frequencies in cord blood and peripheral blood are similar [Immunology. 2011 May; 133(1): 41–50], we only tried the same dose for PB-MNC as we used the dose for CB-MNC. This is exactly what we found intriguing in our study that at the same dose CB-MNC or CB-CD14+ monocytes were neuroprotective compared to the PB-MNC or PB-CD14+ monocytes. 

When we decided to determine whether these cells will be functional even if they are not in direct contact with the brain slices, we added cells below the membrane, into the medium and decided to use 125,000 cells, 5-times more cells than were added directly to the slices, to account for the very large dilution of secreted factors in the culture medium [2µL of medium in meniscus on slice versus 1mL of medium below slice]. Our results show that CB monocytes do secrete protective paracrine factors, which were significantly neuroprotective at 125,000 cells/well dose and we did not test any other doses. To make it more explicit we have now edited the method section [p. 7, line 22-23 and p. 8, line 1-2] and results section (p. 13, line 9) to address these issues.

In the revised manuscript we have also added four new references:

#18: Shahaduzzaman, M.D., et al., Human umbilical cord blood cells induce neuroprotective change in gene expression profile in neurons after ischemia through activation of Akt pathway. Cell Transplant, 2015. 24(4): p. 721-35.

#34: Bachstetter, A.D., et al., Peripheral injection of human umbilical cord blood stimulates neurogenesis in the aged rat brain. BMC Neurosci, 2008. 9: p. 22.

#35: Scotland, P., et al., Gene products promoting remyelination are up-regulated in a cell therapy product manufactured from banked human cord blood. Cytotherapy, 2017. 19(6): p. 771-782.

#52: Shiao, M.L., et al., Immunomodulation with Human Umbilical Cord Blood Stem Cells Ameliorates Ischemic Brain Injury - A Brain Transcriptome Profiling Analysis. Cell Transplant, 2019: p. 963689719836763.

---

## [Editor Report · Decision Letter 1]

14 Aug 2019

Human Umbilical Cord blood monocytes, but not adult blood monocytes, rescue brain cells from hypoxic-ischemic injury: Mechanistic and therapeutic implications

PONE-D-19-16233R1

Dear Dr. Saha,

We are pleased to inform you that your manuscript has been judged scientifically suitable for publication and will be formally accepted for publication once it complies with all outstanding technical requirements.

With kind regards,

Cesar V Borlongan

Academic Editor

PLOS ONE

Additional Editor Comments (optional):

The authors have fully addressed the minor suggestions recommended by both reviewers. This revised manuscript is now prime time for publication. -Cesar V Borlongan
---

## [Editor Report · Acceptance letter]

21 Aug 2019

PONE-D-19-16233R1 

Human Umbilical Cord blood monocytes, but not adult blood monocytes, rescue brain cells from hypoxic-ischemic injury: Mechanistic and therapeutic implications 

Dear Dr. Saha:

I am pleased to inform you that your manuscript has been deemed suitable for publication in PLOS ONE. Congratulations! Your manuscript is now with our production department. 

With kind regards,

on behalf of

Prof. Cesar V Borlongan 

Academic Editor

PLOS ONE